# Using Whole Slide Gray Value Map to Predict HER2 Expression and FISH Status in Breast Cancer

**DOI:** 10.3390/cancers14246233

**Published:** 2022-12-17

**Authors:** Qian Yao, Wei Hou, Kaiyuan Wu, Yanhua Bai, Mengping Long, Xinting Diao, Ling Jia, Dongfeng Niu, Xiang Li

**Affiliations:** 1Department of Pathology, Key Laboratory of Carcinogenesis and Translational Research (Ministry of Education), Peking University Cancer Hospital and Institute, Beijing 100142, China; 2PingAn Technology, Beijing 100016, China

**Keywords:** breast cancer, HER2, artificial intelligence, deep learning, immunohistochemical (IHC) scoring

## Abstract

**Simple Summary:**

HER2 expression is important for target therapy in breast cancer patients, however, accurate evaluation of HER2 expression is challenging for pathologists owing to the ambiguities and subjectivities of manual scoring. We proposed a deep learning framework using a Whole Slide gray value map and convolutional neural network model to predict HER2 expression level on immunohistochemistry (IHC) assay and predict HER2 gene status on fluorescence in situ hybridization (FISH) assay. Our results indicated that the proposed model is feasible for predicting HER2 expression and gene amplification and achieved high consistency with the experienced pathologists’ assessment. This unique HER2 scoring model did not rely on challenging manual intervention and proved to be a simple and robust tool for pathologists to improve the accuracy of HER2 interpretation and provided a clinical aid to target therapy in breast cancer patients.

**Abstract:**

Accurate detection of HER2 expression through immunohistochemistry (IHC) is of great clinical significance in the treatment of breast cancer. However, manual interpretation of HER2 is challenging, due to the interobserver variability among pathologists. We sought to explore a deep learning method to predict HER2 expression level and gene status based on a Whole Slide Image (WSI) of the HER2 IHC section. When applied to 228 invasive breast carcinoma of no special type (IBC-NST) DAB-stained slides, our GrayMap+ convolutional neural network (CNN) model accurately classified HER2 IHC level with mean accuracy 0.952 ± 0.029 and predicted HER2 FISH status with mean accuracy 0.921 ± 0.029. Our result also demonstrated strong consistency in HER2 expression score between our system and experienced pathologists (intraclass correlation coefficient (ICC) = 0.903, Cohen’s *κ* = 0.875). The discordant cases were found to be largely caused by high intra-tumor staining heterogeneity in the HER2 IHC group and low copy number in the HER2 FISH group.

## 1. Introduction

Breast cancer is the most diagnosed cancer that seriously threatens the life and health of women all over the world, with high morbidity and mortality rates of 24.5% and 15.5%, respectively [1]. The HER2 (human epidermal growth factor receptor-2) gene, located at chromosome 17q12–21^2^, plays an important role in the development of breast cancer. Fifteen to twenty percent of breast cancer patients are HER2 positive, including HER2 gene amplification and/or overexpression. HER2-positive breast cancer has poor clinical outcomes [2,3], but fortunately, there is a targeted drug-Trastuzumab (Herceptin), which can effectively improve the prognosis [4,5]. HER2 gene amplification assessed by in situ hybridization (ISH) or protein overexpression assessed by IHC remains the primary predictor of responsiveness to HER2- targeted therapies and a key prognostic biomarker in breast cancer [6]. According to the latest American Society of Clinical Oncology (ASCO)/College of American Pathologists (CAP) guideline [6], all newly diagnosed patients with breast cancer must have a HER2 test performed. In routine clinical practice, the IHC test is first performed. The IHC test gives a score of 0, 1+, 2+, or 3+ that measures the amount of HER2 receptor protein on the surface of cells in a breast cancer tissue sample. The 3+ is the strongest staining, with which the patient must be diagnosed as HER2 positive. 2+ is also known as the equivocal level. Fluorescence in situ hybridization (FISH) must be performed to further decide the HER2 status for patients with IHC 2+ score. Therefore, accurate and efficient HER2 IHC evaluation is important for the diagnosis and treatment of breast cancer patients. In the HER2 IHC test, the HER2-receptor protein is commonly stained with 3,3′-diaminobenzidine (DAB), which has a brown color, meanwhile, hematoxylin staining which has blue color is also applied to visualize the cell nuclei. The stained slide is manually accessed by pathologists under the microscope. Although many countries have implemented national testing guidelines to standardize testing procedures and make results more accurate, the procedure is subjective and semi-quantitative and quite often leads to high inter- and intra-observer variation [7,8,9]. Therefore, there is an urgent need for an objective and consistent HER2 evaluation system.

Many researchers are devoted to developing computer-aided solutions, semi-automatically or fully automatically, to address the ambiguities and subjectivities of manual scoring. Compared to manual scoring, the computer-aided solution can decrease human error, increase the accuracy of diagnosis, reduce the workload of pathologists, and standardize the scoring systems [10,11]. The pathology whole slide images (WSI) have trillions of pixels, which are too large to process in a single-shot end-to-end way, i.e., processing WSI as a traditional image, even on modern computers. Usually, the fully automatic methods have the following three steps: WSI is first split into small size, i.e., 512 × 512, image patches; then information of single patch image are extracted; and at last single patch information are summarized to conclude the WSI level result. While the semiautomatic methods need pathologists to manually select regions of interest in the WSI. Masmoudi, et al. [12] presented a method for automated assessment of HER2 IHC staining. They first used a linear classification model on the color information of pixels to discriminate the membrane pixels and nuclei pixels, then watershed algorithm and adaptive ellipse fitting were applied to segment the nuclei and cell membrane. At last, slides were classified into one of the three scoring groups based on features describing the membrane staining intensity and completeness. In contrast to Masmoudi et al. work, HER2CONNECT found the distribution of the area of the connected brown color components (the stained membranes) in the core invasive cancer region had a good correlation with the HER2 expression level, therefore can be used to predict HER2 score. Their method reached 92.3% between the software and the score by the pathologist [13]. Ruifrok et al. [14] proposed a color deconvolution method to deconvolute and quantify the contributions of each staining in the histochemical slide. Motivated by the color convolution method, many researchers were devoted to quantifying the gray level of the HER2 IHC slide. ImmunoMembrane, a web-based application, utilized color deconvolution to separate stained membranes and then designed the IM-score, which is the sum of membrane completeness score and membrane intensity score to classify HER2 scores [15]. Kabakci et al. [16] characterized the cell membrane staining intensity in a comprehensive way using the so call Membrane Intensity Histogram (MIH) method which described the distribution of the staining intensity in different directions.

Deep Learning (DL) models are increasingly being used in various application areas such as computer vision, natural language processing, text or image classification, sentiment analysis, recommender systems, user profiling, etc. [17,18]. Compared to handcraft feature engineering, one of the major advantages of the DL model is the automatic learning feature representation and high representability, which bring the DL model much more versatility when dealing with large datasets and complex problems. Saha et al. [11] developed a cell segmentation model using Trapezoidal LSTM units and HER2 scoring based on the segmented membranes. However, Saha uses 2048 × 2048 patches, rather than the entire WSI. Qaiser et al. [19] also achieved patch-level HER2 scoring with the help of reinforcement learning. Zhen Chen, et al. [20] proposed a Focal-Aware Module to estimate diagnosis-related regions and a Relevance-enhanced Graph Convolutional Network to summarize information extracted from different levels of the original WSI.

Recently DL models are attracting increasing attention to predicting gene expression status using the WSI image [21,22,23,24]. The diagnosis label is usually provided at the WSI level, which cannot be treated as a cluster label of the inputs of the underline model. Therefore, multiple instance learning (MIL) is often implemented to overcome the issue. In this paper, we propose a new artificial intelligence (AI) method to predict HER2 protein expression level and gene status using the WSIs. Instead of using a manual strong label of patch level image or using MIL on the slide-level labeled dataset, we first calculate the unsupervised feature for each patch image, i.e., the gray level, the gray level area fraction, and generate a slide-level feature map using the patch-level feature to represent each patch. In this way, we can reduce the input size of the original slide. Then we build a multi-task deep learning model to predict HER2 protein expression level and gene amplification status simultaneously.

## 2. Material and Methods

Figure 1 shows the workflow of our study.

### 2.1. Human Subjects

We selected 228 biopsy cases of IBC-NST with both IHC and FISH information which were collected between 2010 and 2021 from the department of pathology, Peking University Cancer Hospital & Institute. All subjects were female. Our study obtained permission from the Peking University Cancer Hospital Institutional Review Board and Ethics Committee (Grant: 2022KT15).

### 2.2. ImmunohistoChemical Staining

Commercially available primary antibody HER2 (4B5, Roche Ventana) was applied. Immunohistochemical stains were performed on Ventana Benchmark automated immune-Stainer (Tucson, Arizona), following the vendor’s protocol. The appropriate positive and negative controls were included for each run. HER2 immunoexpressing was evaluated as 0, 1+, 2+, and 3+ based on the 2018 ASCO/CAP guideline [6] by three experienced pathologists (Q.Y., D.N., and Y.B.). To prevent intra-rater variability, three pathologists were blind to the initial manual evaluation and AI-based scores, and all the cases were reviewed a second time after a 4-week washout period. The discrepant cases were reviewed again to get the final score.

### 2.3. Fluorescence In Situ Hybridization

HER2 FISH was carried out using the Path Vysion HER2 DNA Probe Kit (Abbott Molecular, Abbott Park, Illinois) and followed the manufacturer’s instructions. Two experienced pathologists (DFN and Y.B.) evaluated the HER2 copy number, CEP17 copy number, and their ratios of 20 tumor cells independently and blinded to IHC results. FISH results were recorded as negative and positive according to the 2018 ASCO/CAP guideline. In detail, HER2 FISH results were designated into five groups: group one (G1, HER2/CEP17 ratio ≥ 2.0; average HER2 copy number ≥ 4.0/cell); group two (G2, HER2/CEP17 ratio ≥ 2.0; average HER2 copy number < 4.0/cell); group three (G3, HER2/CEP17 ratio < 2.0; average HER2 copy number ≥ 6.0/cell); group four (G4, HER2/CEP17 ratio < 2.0; 4.0 ≤ average HER2 copy number < 6.0/cell); and group five (G5, HER2/CEP17 ratio < 2.0; average HER2 copy number < 4.0/cell) [6]. G1 was considered FISH positive and G5 was FISH negative. However, G2 and G4 should evaluate the HER2 IHC results in addition, if not 3+, then those cases should be considered HER2 negative. In G3 cases, when concurrent IHC results are negative (0 or 1+), it is recommended that the specimen be considered HER2 negative.

### 2.4. Image Processing

The digitized whole-slide images (WSIs) were acquired using a Leica Aperio Versa pathologic scanner (Aperio, Leica Biosystems Imaging, Inc.) viewed at 400× magnification using Leica ImageScope software. The order of magnitude of pixels was $10^{9}\sim 10^{10}$.

Figure 1 shows the flowchart of the method. The whole slide image was first partitioned into 512×512 patches. Then for each small patch image, we segment the membrane pixels using color deconvolution and the k-means method (k-means parameters: number of clusters is 3, the maximum number of iterations is 50, number of redos is 10). After the membrane segmentation, we evaluate the gray value and membrane pixels fraction of each patch. The original WSI is profiled into three maps. In the following, we describe the procedure in detail.

### 2.5. Membrane Segmentation

The DAB signal is mainly located at the membrane. In the following, we introduce the membrane segmentation method which is based on the color deconvolution and k-means method. Ruifrok etc. applied the Beer-Lambert law to model the stained slide image and proposed the color deconvolution method to separate and quantify immunohistochemical staining [14]. According to the Beer-Lambert law,
(1)$$I_{c}=I_{0,c}10^{-AC_{c}}$$
where $I_{c}$ is the intensity of light detected after passing the specimen, $I_{0,c}$ is the intensity of light entering the specimen and *A* is the amount of the stain with absorption factor *C*. The subscript *c* indicates the detection channel. By assuming a linear relation between stain concentration and absorbance, Ruifrok proposed the following color deconvolution method,
(2)$$A=-\log10\left(\frac{I}{I_{0}}\right)\times OD^{-1}$$
where A is a vector representing the amount of different stains, I is the transmitted light intensity, i.e., the detected slide image, OD is the normalized optical density matrix, which can be measured experimentally. In the analysis of the HER2 IHC slide, because there are only two kinds of stains, we use the following normalized *OD* matrix
(3)$$OD=\left(\begin{array}{ccc} 0.650 & 0.704 & 0.286\\ 0.268 & 0.570 & 0.776\\ 0.636 & -0.710 & 0.302 \end{array}\right)$$
where the first two row vectors correspond to the *OD* vectors of hematoxylin and DAB^14^ and the last row vector is the normalized cross product of hematoxylin and DAB *OD* vectors. Following the convention of color deconvolution code given in the Color Deconvolution 2

ImageJ plugin, we use $A=-\log10\left(\frac{I}{255}\right)\times OD^{-1}$ to deconvolute the original slide image.

After color deconvolution, the value of the 2nd channel corresponds to the intensity of the DAB stain. We then apply the k-means method to the original image. The image is first converted from RGB to Luv color to get better perceptual uniformity which is more suitable for clustering analysis. Define the distance between pixels p,q:(4)$$D\left(p,q\right)=\sqrt{{\left(L_{p}-L_{q}\right)}^{2}+{\left(u_{p}-u_{q}\right)}^{2}+{\left(v_{p}-v_{q}\right)}^{2}}$$
where $\left(L_{p},u_{p},\;v_{p}\right)$ and $\left(L_{q},\;u_{q},v_{q}\right)$ are Luv values of pixel *p* and *q*, respectively. Based on the distance D(p,q), we use the k-means algorithm to cluster the pixels in the slice into three clusters, which correspond to the stained cell membrane region, the nuclei region, and the complementary region respectively. At last, we calculate the mean gray values of each pixel group according to the DAB channel calculated previously. We select the group with the highest mean gray value as the cell membrane. Figure 2A–D gives an illustration of the cell membrane segmentation.

### 2.6. Gray Value Map

In this section, we describe the gray value map which integrates patch-level gray value information to get slide-level gray value information. After segmentation of the cell membrane of each patch image, we calculate the mean gray value and membrane pixel fraction of each patch image. We find that the value of the DAB channel cannot reflect well when the visual gray value is greater than 8, as shown in Figure 2E. By checking the RGB channel value of the membrane pixels, we find that this effect is partially caused by the saturation of the blue channel. It is unclear whether this is truly caused by the stain absorbing all blue light or whether there are some other effects of the hardware device. We notice that the Lightness channel of Luv color space generally reflects the visual gray level except the low gray value range. Therefore, we add the Lightness channel value to the gray value map and build the model to automatically fuse the information. In summary, the gray value *A*, membrane pixel fraction *F,* and Lightness value *L* at patch level are defined as:

$A={\mathrm{mean}}_{i}A_{i}$ where mean is over all pixels in the membrane cluster,

$F=\frac{\mathrm{number}\;\mathrm{of}\;\mathrm{pixels}\;\mathrm{in}\;\mathrm{membrane}\;\mathrm{cluster}}{\mathrm{total}\;\mathrm{number}\;\mathrm{of}\;\mathrm{pixels}}$,

$L={\mathrm{mean}}_{i}L_{i}$ where mean is over all pixels in the membrane cluster.

Figure 3 shows the gray value map of IHC HER2 expression 0/1+, 2+, and 3+ cases.

### 2.7. Multitask Convolutional Neural Network (CNN)

After getting the gray value map of the whole slide, we further utilize a multi-task CNN model to classify the IHC HER2 expression level and the FISH status simultaneously. We use Resnet18 with base channel number 64 as our backbone network. After the backbone network, we concatenate two task branches corresponding to the IHC HER2 expression classification and the FISH status classification respectively. For each task branch, we use the sigmoid cross-entropy loss as the classification loss and add the dropout layer before the last fully connected layer. All Relu activations are replaced with PRelu to avoid the Relu blow-up issue due to a lack of pretrained weight initialization.

Data augment techniques and manually synthesized images are used to overcome the overfit issue due to the lack of training data samples. We add random rotation (−180, +180), random crop (512, 512) (raw training input size is (680, 680)), random horizontal flip, and random vertical flip data augmentations. We also manually synthesize the image for each original data sample by first manually drawing a mask of a random sample that has the same FISH status, and the same fold-id, but a lower HER2 expression level of the target sample, and then paste the masked part of the selected sample into the target sample’s blank space. In this way, we partially increase our training dataset.

The model is implemented in Pytorch using the MMDetection framework and trained with the Adam optimizer with Cosine learning rate policy (learning rate parameters: base learning rate is 0.001, the minimum learning rate is 1.0 × 10^−8^). We utilized the 5-fold cross-validation method to evaluate the model. The mean and standard deviation were calculated using prediction on each fold to demonstrate the model performance and stability. Evaluation metrics including precision, recall, F1-score, Jaccard Index, specificity, accuracy, and Area Under Curve of receiver operating characteristic curve (ROC) (AUC) were calculated for binary FISH status prediction. Evaluation metrics including accuracy, F1-score, Cohen’s kappa coefficient (κ), and Matthews correlation coefficient (MCC) were calculated for multiclass IHC prediction using macro average mode.

## 3. Results

### 3.1. HER2 IHC Status Classification Using GrayMax Model

In the first step, we obtained the manual results of HER2 IHC and HER2 FISH. HER2 IHC was evaluated by three experienced pathologists. We used the median score of three pathologists to further reduce the inter-observer variability, which meant if there was a difference between the three scores, we used the median value of three scores. The details of the HER2 status including IHC and FISH results are shown in Table 1. According to the 2018 ASCO/CAP clinical practice guideline, the cutoff of HER2 IHC staining is 10%, which means the 10% strongest staining of HER2 IHC can be chosen as the represent score of the whole slice. So, we first use the maximum gray value of all patches to represent the gray value of WSI. Then we compared the GrayMax model with the median HER2 scores of pathologists. However, after utilization of the 5-fold cross-validation method, the GrayMax model showed relatively inferior performance with an average accuracy of 0.842±0.023, F1-score of 0.665±0.078, *Cohen’s* κ of 0.640±0.063 and MCC of 0.663±0.058 (Table 2). We analysed the details of our model and found the errors in the cases with a heterogeneity of staining, nonspecific cytoplasmic staining, and in cases with invasive micropapillary carcinoma component, mucinous carcinoma component and ductal carcinoma in situ (DCIS) component and interference by necrosis region.

### 3.2. HER2 IHC Status Classification Using GrayMap + CNN Model

To solve the issues of the GrayMax model, we developed a new method to classify the HER2 IHC status. The main issue of the GrayMax model is that a single maximum gray value cannot represent the information of the whole slide. Therefore, we first used the GrayMap of the original whole slide, which contained the gray value information of all the patches, as described in the materials and methods section. Figure 2 showed the segmentation of the cell membrane and the schematic of GrayMap. Figure 3 showed typical examples of GrayMap in a subgroup of 0/1+, 2+, and 3+. Next, we utilized a multi-task CNN model to classify the IHC HER2 expression level as described in the material and methods section (Figure 1). We evaluated the model through a 5-fold cross-validation method and compared the results with three experienced pathologists. The experiment results show that the GrayMap model has much better performance than the GrayMax model with an average accuracy of 0.952±0.029, F1-score of 0.860±0.12, *Cohen’s* κ of 0.891±0.069 and MCC of 0.899±0.062 (Table 2). Parameters of evaluation metrics on a subgroup of 0/1+, 2+, and 3+ showed in Figure 4A and Appendix A. We further analyzed the intraclass correlation coefficient (ICC) among pathologists and found the ICC value was 0.791 (95% confidence interval [CI], 0.749–0.829) (Figure 4B). It indicated the presence of inter-observer variability and suggested that manual interpretation by the single pathologist may face a high risk of misdiagnosis. Then HER2-AI and HER2-pathologists were compared to show consistency between the AI system and pathologists. The median variables of HER2 pathologists were used in the comparison. The results showed a high consistency between the HER2-AI and HER2-pathologists (ICC = 0.903) (Figure 4C).

### 3.3. HER2 Gene Status Prediction Using GrayMap+ CNN Model

Since HER2 IHC expression largely represents the HER2 gene amplification status [25]. We also utilized the GrayMap model to predict HER2 gene status and compared the data with the FISH results. Our system demonstrated high performance in predicting HER2 gene status with an accuracy of 0.921, specificity of 0.945, precision of 0.927, recall of 0.89, F1-score of 0.908, and Jaccard Index of 0.832 (Figure 5A and Appendix A) and AUC value of 0.936 in the ROC curve which presented the high quality in FISH classification via 5-fold cross-validation method (Figure 5B). This data further confirmed our model as a robust high-performance system not only in HER2 IHC classification but also in HER2 gene status prediction.

### 3.4. The Analysis of Discordant Cases

The proposed system correctly classified most of the WSIs. However, there were several discordant cases with false positive and negative samples (Figure 6A). We further analyzed the difference between AI systems and pathologists. As for the HER2 IHC results, 13 (13/228, 5.70%) cases were discordant between AI and pathologists. We investigated each case to identify the causes of the variability. Intra-tumor cell heterogeneity of HER2 staining was detected in six cases (6/13, 46.15%) (Figure 6B). Nonspecific cytoplasmic staining was found in four cases (Figure 6C). Another one was due to the nonspecific staining in DCIS (Figure 6D). Our result provided that HER2 staining heterogeneity was identified as the main driver of disagreement between AI and pathologists. Furthermore, the cytoplastic staining can interfere with the machine’s extraction of cell membrane staining, resulting in misinterpretation. The nonspecific HER2 expression on DCIS will also lead to error, especially on biopsy tissue with a substantial amount of DCIS. HER2 validation is supposed to be performed only in the IBC-NST component. Since we did not annotate the IBC-NST region on WSIs, we calculated the DCIS component and found 75 cases (75/228, 32.89%) of samples had a DCIS component with a ratio of 5–35%. Only one case (1/75, 1.33%) was included in discordant cases, thus, our model had the ability to resolve the hidden trouble of DCIS. Only two cases could not find a clear explanation for discordance. According to HER2 FISH status, there were 18 (18/228, 7.89%) discordant cases. Five cases were identified intra-tumor cell heterogeneity of dual-color probes. For example, one case with only 2% tumor cells HER2 amplification and one case with 5%. Seven cases have low HER2 copy numbers (average copy number range 4–6 signals/cell). Three cases that were manually evaluated as negative belonged to the G2 and G4 groups, which were the new FISH group according to the 2018 ASCO/CAP guideline. Though the seven low-copy number cases were evaluated as positive and the new FISH group was regarded as negative, the efficacy of HER2-targeted therapy on these groups still needs to be investigated because of the limited evidence with a small subset of cases [6]. Only five cases were left without any explanation for discordance. Our results indicated that AI-based classification guaranteed high diagnostic accuracy and enabled us to reduce misinterpretation.

## 4. Discussion

In this paper, we proposed a new AI method to tackle the subjectivity and inter-observer disagreement issues of manual interpretation of HER2 IHC slides. The experiments’ results showed that the new method could accurately predict HER2 protein expression level (Accuracy 0.95±0.029, Cohen’s κ 0.891±0.069) and FISH status (AUC 0.936±0.030). The test of concordance with the three pathologists’ interpretation showed that the new method has the highest ICC (ICC 0.903, 95%-Confidence Interval 0.875~0.924). Breast cancer (BC) has become the most common cancer diagnosed in women. Personalized medicine, especially drugs focused on target genes in BC, such as trastuzumab, has greatly improved survival. HER2 protein expression level and gene amplification status are the most important indicators for the targeted therapy of BC. However, traditional manual interpretation of HER2 slide has been criticized for subjectivity and inter-observer disagreement among pathologists. This is not only caused by the subjective decision that needs clinic pathologists to take, such as completeness of the membrane staining, intensity of staining, and percentage of positive cells, according to the ASCO/CAP guideline, but also caused by the heterogeneity of BC. AI-based methods, because of the nature of the parametrized model and deterministic behavior, are a prospective approach to solving the pool reproducibility issue of manual interpretation. However, on one hand, the whole slide image is too large to be processed by a single model directly, on the other hand, a single patch-level image of WSI is not able to capture the heterogeneity property of BC. Currently, there are several approaches to solving this issue. The first approach predicts the HER2 expression of each patch and uses the statistical average method to summarize the patch-level results. Compared to this approach, the method proposed in this work adopts a deep learning model to do slide-level predictions, which are more flexible and powerful than the simple statistical average method. Another approach generally follows the ASCO/CAP guideline, making predicting at the cell level. This approach needs considerable human labeling which is not only tedious but also prone to label error, especially for weak staining samples. The weakly Supervised Learning (WSL) method is an attractive method to alleviate patch-level labeling [26]. However, WSL needs a considerable amount of slide-level data. Currently, the performance of WSL on a large HER2 IHC dataset is unclear yet. The method proposed in this work could be another prospective approach to do slide-level predictions.

The proposed AI system can be applied in our actual work in the pathology department. After uploading the WSIs into the system, our model can automatically process patches splitting, cell segmentation, gray value map information extraction, and HER2 IHC and FISH results prediction. The system assists pathologists by pre-reading HER2 IHC slides and presenting calculated results as second opinions to pathologists, especially those with equivocal results as 2+. Our system will significantly mitigate the interobserver discrepancy and contribute to the efficacy and safety of HER2-targeted therapies on BC. At present, a new HER2-low subtype was defined by a score of IHC 1 +or IHC 2+/FISH −, who may benefit from the new HER2-ADC drugs, such as trastuzumab deruxtecan (T-DXd) [27]. The current system has the potential to recognize HER2-low cases with an accurate prediction of both IHC and FISH status.

In our study, compared to the former GrayMax algorithm, the upgraded GrayMap + CNN model can get rid of the most nonspecific and heterogeneous staining problem as well as the special staining pattern of specific breast cancer subtypes in HER2 IHC classification. However, inconsistency between AI systems and pathologists still exists. Consistent with the previous study, HER2 staining heterogeneity was identified as the main driver of disagreement [28]. Intratumoral heterogeneity of HER2 may be due to intrinsic the characteristics of BC, defined as regional heterogeneity and genetic heterogeneity [29]. It may also be caused by IHC procedures, tissue collection, and processing, or slide scanning procedure. In our dataset, most heterogeneity staining cases of the discordant cohort were weak staining thus our model need to improve its capability in dealing with weak HER2 staining. As for HER2 FISH classification, in addition to heterogeneity, a low copy number (average copy number range 4–6 signals/cell) was the most common cause of inconsistency. According to the 2018 guideline, an average HER2 copy number ≥4 signal/cell is regarded as FISH positive. However, the study showed a clear difference on HER2 copy levels using droplet digital PCR (ddPCR) and targeted next-generation sequencing (NGS) method between the 4–6 copy number groups and ≥6 groups. However, it remains unclear if patients of the 4–6 copy number group derive the same level of benefit as the≥6 groups in HER2-targeted therapy [30]. Futhermore, there were three cases belonging to G2 and G4 groups according to the 2018 guideline, which was the new FISH and should be recognized as FISH negative. However, the researcher showed the G2 group represents a biologically heterogeneous subset, which is different from those in G1 (FISH positive) and G5 (FISH negative) [31]. The G4 group was also proved to be a distinct group with intermediate levels of RNA/protein expression, close to positive/negative cut points [32]. Additional outcome information after HER2-targeted treatment is needed for the new FISH groups.

To improve the accurate, precise, and reproducible interpretation of HER2 IHC results for BC, where quantitative image analysis (QIA) is applied, The College of American Pathologists (CAP) developed the guideline with eleven recommendations [33]. The recommendations suggested that QIA and procedures must be validated before implementation, followed by regular maintenance and ongoing evaluation of quality control and quality assurance. In addition, HER2 QIA performance, interpretation, and reporting should be supervised by pathologists with expertise in QIA. We studied the detailed description of the guideline and found our AI model and procedures met most of the criteria, which suggested the present model is a promising tool for HER2 interpretation. However, this study still had some limitations. First, this work uses the k-means method to segment the cell membrane. It may wrongly classify the cytoplasmic pixels into membrane when the cell is weakly stained or cytoplastic immunohistochemical staining. For most of the weakly stained cases, the method is still able to do correct predictions, because the intensity and percentage of positive cells are major discrimination factors. However, for cytoplastic staining cases, as also demonstrated in the analysis of discordant cases section (four out of 13 total error cases), more local features are needed to discriminate the wrong cases. Secondly, we did not segment the invasive carcinoma region first. The current method relies on the deep learning model to automatically learn features from the data. In future works, we will collect more data and investigate the performance difference between the current method and model which makes predictions only rely on carcinoma region. Third, the completeness of the cell membrane is not represented in the current method. 2018 ASCO/CAP guidelines lay more emphasis on the completeness of cell membrane staining on HER2 2+ and 3+ cases in order to reduce the confusion of pathologists and allow greater discrimination between positive and negative results [6]. Our AI system promised high performance without calculating membrane completeness, however, a feature still needed to be found to represent the completeness of cell membrane staining according to the ASCO/CAP guideline to get a better result.

In conclusion, experimental results indicated that the proposed AI model is feasible for predicting HER2 expression score and HER2 gene amplification using IHC WSI and achieved high consistency with the experienced pathologists’ assessments. This unique HER2 scoring model does not rely on challenging manual intervention and is proven to be a simple and robust tool for pathologists to improve the accuracy of HER2 interpretation and provides a clinical aid to target therapy in BC patients.

## Figures and Tables

**Figure 1 cancers-14-06233-f001:**
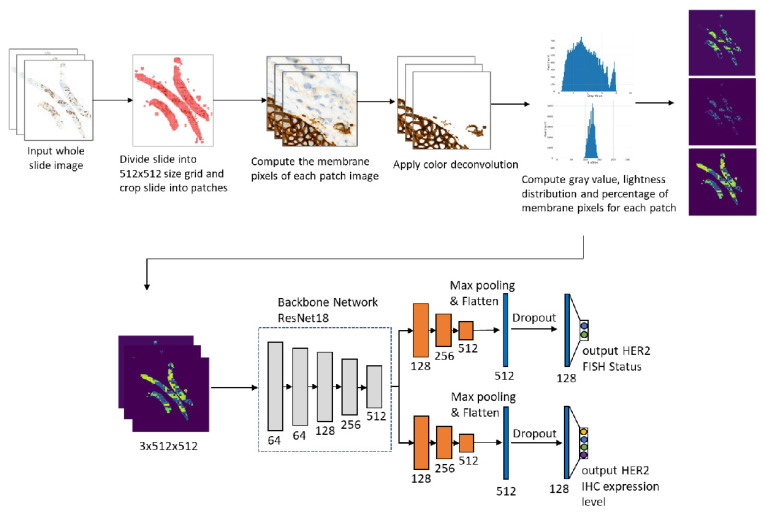
The workflow of our study includes the main steps for preprocessing slides and training the deep learning model. The numbers below the model block give the channel number respectively.

**Figure 2 cancers-14-06233-f002:**
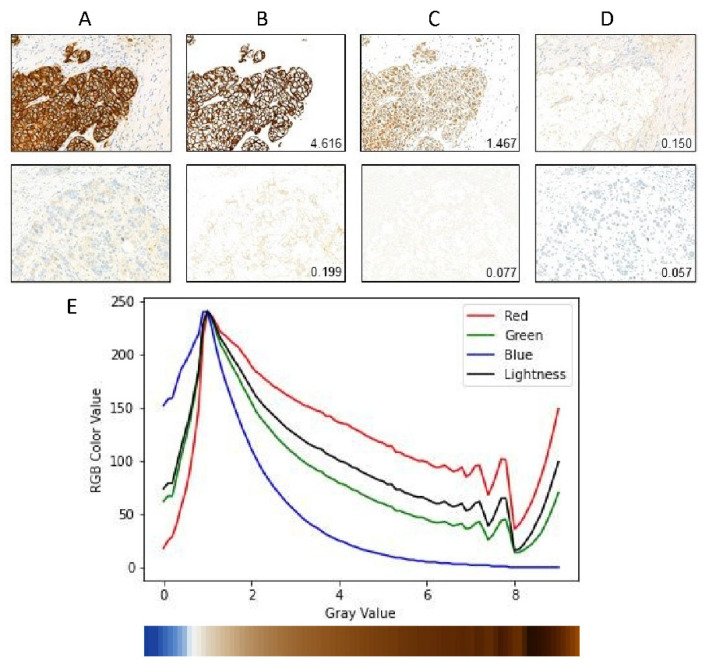
Cell membrane segmentation and the schematic of Graymap. (**A**) raw section of HER2 3+ and HER2 0/1+. (**B**–**D**) are three groups of K-Means output. The gray values are labeled on the images respectively. (**E**) The mean RGB value of different gray value membrane pixels. The bottom color bar is an RGB color map of different gray values.

**Figure 3 cancers-14-06233-f003:**
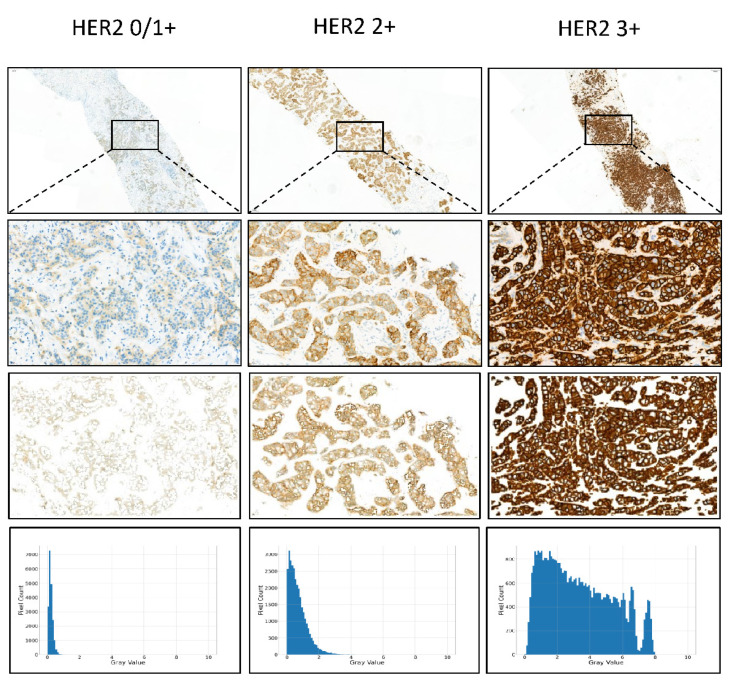
Examples of GrayMap of HER2 IHC expression. Typical examples of HER2 0/1+, 2+, 3+ cases in IBC-NST. From top to bottom: HER2 IHC raw images, magnified images, cell membrane segmentation, and pixels’ gray value’s distribution of the images.

**Figure 4 cancers-14-06233-f004:**
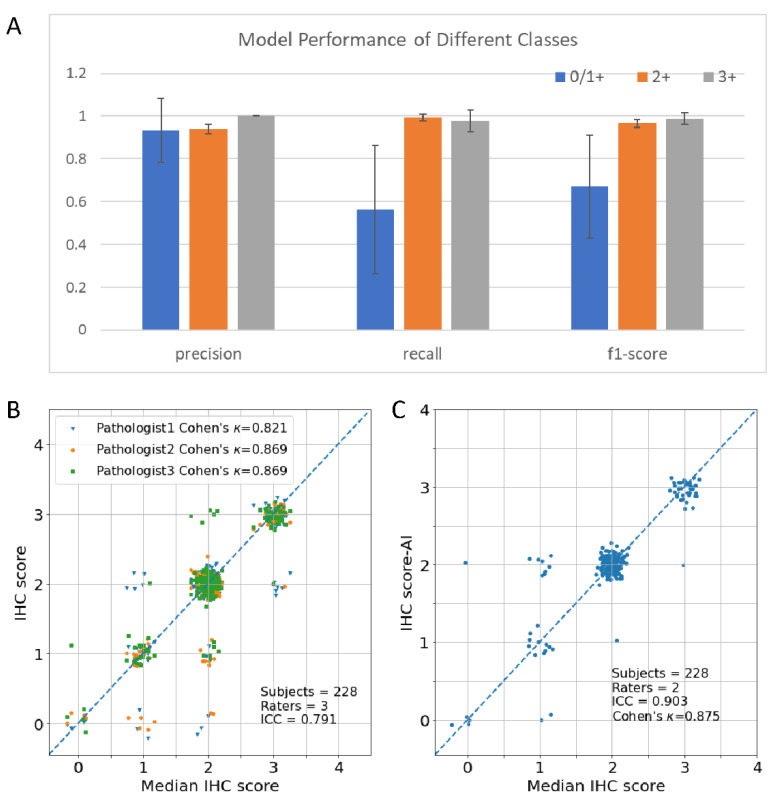
Consistency of the pathologists and the AI system on HER2 IHC classification. (**A**) Histograms of GrayMap model performance in a subgroup of 0/1+, 2+, and 3+. (**B**) The intraclass consistency of HER2 IHC scores in pathologists. (**C**) Consistency of HER2 between AI system (IHC score-AI) and median IHC score in pathologists (median IHC score).

**Figure 5 cancers-14-06233-f005:**
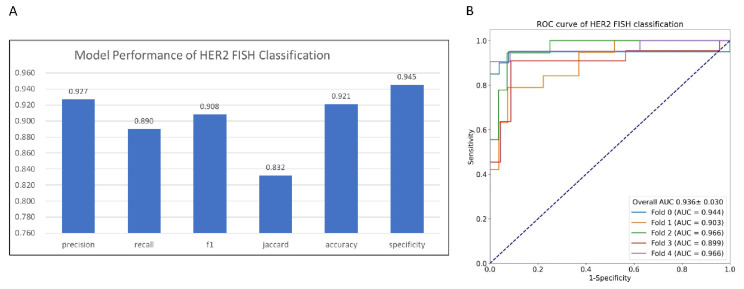
Performance of AI system on HER2 FISH classification. (**A**) Histograms of GrayMap model performance. (**B**) ROC curve of HER2 FISH status classification by cross-validation classification.

**Figure 6 cancers-14-06233-f006:**
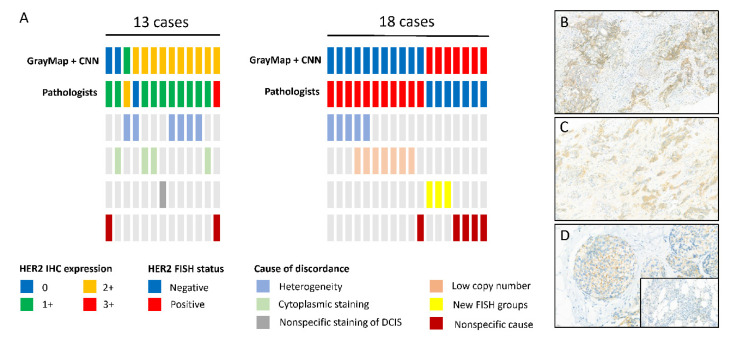
HER2 scoring discordance between pathologists and AI system and the possible causes of the variability. (**A**) Top 2 lines: Comparison between GrayMap model and the pathologist assessment; Bottom 4 lines: The possible causes of the variability; Left: The discordant cases on HER2 IHC classification; Right: The discordant cases on HER2 FISH classification. Vertical bars represent single cases and the representation of different colors are listed at the bottom. The typical image of (**B**) HER2 staining heterogeneity, (**C**) nonspecific cytoplasmic staining, (**D**) nonspecific staining in ductal carcinoma in situ (DCIS) with negative staining of the invasive component.

**Table 1 cancers-14-06233-t001:** Summary of the cohort of the different HER2 statuses.

HER2 Expression Score					
Fish Status	*n*	0	1+	2+	3+
Negative	128	5	19	104	0
Positive	100	0	2	53	45

**Table 2 cancers-14-06233-t002:** Performance comparison of GrayMax and GrayMap + CNN methods by cross-validation classification.

Method	Fold	Accuracy	F1	Kappa	MCC
GrayMax	0	84.78%	69.71%	67.83%	68.51%
	1	84.78%	70.79%	60.92%	67.47%
	2	86.96%	76.87%	72.23%	73.87%
	3	80.00%	55.38%	53.71%	56.18%
	4	84.44%	59.93%	65.27%	65.49%
	Avg.	84.19%	66.54%	63.99%	66.30%
	Std.	2.28%	7.78%	6.31%	5.77%
GrayMap + CNN	0	93.48%	63.63%	83.13%	84.38%
	1	91.30%	84.65%	80.55%	82.54%
	2	95.65%	94.13%	91.54%	91.96%
	3	100.00%	100.00%	100.00%	100.00%
	4	95.56%	87.81%	90.36%	90.36%
	Avg.	95.20%	86.04%	89.12%	89.85%
	Std.	2.88%	12.39%	6.86%	6.18%

Abbreviation: Avg, Average value; Std, Standard deviation.

## Data Availability

All image data associated with this study can be downloaded at https://data.mendeley.com/datasets/3njjk252vc/draft?a=29e5963c-e2d6-4bdb-9c7b-b51d3741b6f0. The source code and the guideline are publicly available at https://github.com/KaiyuanWu/Her2GrayMap. Any further information and requests for resources and materials should be directed to and will be fulfilled by the lead contact, Dongfeng Niu (dongfengniu@foxmail.com).

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
