# Peer review of "Using Whole Slide Gray Value Map to Predict HER2 Expression and FISH Status in Breast Cancer"

_cancers, 2022, doi:10.3390/cancers14246233_

Round 1
Reviewer 1 Report
Introduction
l There are too many spaces before the second paragraph of this section "Many researches are devoted..." . Please, revise it.
l Some specific data can be added to illustrate breast cancer is the most diagnosed cancer with high mortality thus seriously threat ens the life and health of women all over the world.
Supplementary Materials
l "Table S1" should be bold as "Table S2". Please, revise it.
Result
l Why the cutoff of HER2 IHC staining is 10%, which means the 10% strongest staining of HER2 IHC can be chosen as the represent score of the whole slice?
Discussion
l Changing the last paragraph of this section " This unique HER2 scoring model didn't rely on challenging manual intervention" to simple present tense will be better.
Reviewer 2 Report
The authors proposed a deep learning framework using whole slide gray value map and convolutional neural network model to predict HER2 expression level on immunohistochemistry (IHC) assay and predict HER2 gene status on fluorescence in situ hybridization (FISH) assay. The authors claimed the proposed model is feasible for predicting HER2 expression and gene amplification. Some comments:
· Most of the readers are clinicians and cancer researchers who do not understand the sophisticated algorithms of the model. Figure 1 demonstrated the workflow of their study. People might understand predicting HER2 IHC expression level by computing gray value, lightness distribution, and percentage of membrane pixels. But how could IHC whole slide image predict HER2 FISH status? Was FISH positive or negative from 228 cases the only FISH information used to train the model? What are the definitions of FISH positive and negative? (Page 4 under 2.3 Fluorescence in situ hybridization, last sentence). There are 5 FISH pattern groups defined based on the 2018 ASCO/CAP guideline. Additional workup is usually needed for certain groups in order to determine positivity.
· For Table 1, recommend changing “FISH Status – Negative or Positive” to 2018 ASCO/CAP guideline FISH group 1-5. In that case people will easily see what FISH group(s) is(are) more likely to be involved in discordant situation.
· There is a guideline from the CAP for quantitative image analysis of HER2 IHC for breast cancer published in 2019. The authors should read and incorporate in their discussion.
Bui MM, Riben MW, Allison KH, Chlipala E, Colasacco C, Kahn AG, et al. Quantitative Image Analysis of Human Epidermal Growth Factor Receptor 2 Immunohistochemistry for Breast Cancer: Guideline From the College of American Pathologists. Arch Pathol Lab Med. 2019.
· The authors should discuss whether this model has been used in other cancers by other groups.
